# Correlation of Solubility Thermodynamics of Glibenclamide with Recrystallization and In Vitro Release Profile

**DOI:** 10.3390/molecules27041392

**Published:** 2022-02-18

**Authors:** Ravi Maharjan, Junoh Jeong, Ripesh Bhujel, Min-Soo Kim, Hyo-Kyung Han, Nam Ah Kim, Seong Hoon Jeong

**Affiliations:** 1BK21 FOUR Team, Integrated Research Institute for Drug Development, College of Pharmacy, Dongguk University, Gyeonggi 10326, Korea; raavii@dgu.ac.kr (R.M.); junoh0621@naver.com (J.J.); ripeshbhujel@gmail.com (R.B.); hkhan@dongguk.edu (H.-K.H.); namah87@dongguk.edu (N.A.K.); 2College of Pharmacy, Research Institute for Drug Development, Pusan National University, Busan 46241, Korea; minsookim@pusan.ac.kr

**Keywords:** solubility, glibenclamide, solvate, dissolution

## Abstract

The solubility of glibenclamide was evaluated in DMSO, NMP, 1,4-dioxane, PEG 400, Transcutol^®^ HP, water, and aqueous mixtures (*T* = 293.15~323.15 K). It was then recrystallized to solvate and compressed into tablets, of which 30-day stability and dissolution was studied. It had a higher solubility in 1,4-dioxane, DMSO, NMP (*X*_exp_ = 2.30 × 10^3^, 3.08 × 10^4^, 2.90 × 10^4^) at 323.15 K, its mixture (X_exp_ = 1.93 × 10^3^, 1.89 × 10^4^, 1.58 × 10^4^) at 298.15 K, and 1,4-dioxane (*w*) + water (1−*w*) mixture ratio of *w* = 0.8 (*X*_exp_ = 3.74 × 10^3^) at 323.15 K. Modified Apelblat (*RMSD* ≤ 0.519) and CNIBS/R-K model (*RMSD* ≤ 0.358) suggested good comparability with the experimental solubility. The minimum value of ΔG° vs ΔH° at 0.70 < *x*_2_ < 0.80 suggested higher solubility at that molar concentration. Based on the solubility, it was recrystallized into the solvate, which was granulated and compressed into tablets. Among the studied solvates, the tablets of glibenclamide dioxane solvate had a higher initial (95.51%) and 30-day (93.74%) dissolution compared to glibenclamide reference (28.93%). There was no stability issue even after granulation, drying, or at pH 7.4. Thus, glibenclamide dioxane solvate could be an alternative form to improve the molecule’s properties.

## 1. Introduction

Pharmaceutical scientists relentlessly try to improve the solubility and stability of different solid forms and incorporate the most suitable attributes to improve their therapeutic effect and processability [1]. Among the various solid forms, solvates or polymorphs can be an exciting tool for generic manufacturers who are looking for an alternative solid form to bypass the innovators’ patents. In the solid system, the phase differs in its elemental composition in response to the addition of different solvents. Different unit operations, such as high-shear wet granulation and freeze-drying processes, may affect the stability of the solvate form. Thus, a stability study under different temperature conditions and solvent activity is essential for the development of the dosage form. Only a limited number of drugs in solvate form are commercially available. Therefore, it could be worth exploring different aspects of solvate forms, i.e., the conditions under which the solvate is formed, possible changes in the crystal form during high-shear granulation, drying, and tableting, and the possible benefits from the academic perspectives.

A model drug considered for the present study is glibenclamide (GLN, Figure 1, 5-chloro-*N*-(2-{4-[(cyclohexylcarbamoyl)sulfamoyl]phenyl}xethyl)-2-methoxy benzamide, CAS number: 10238-21-8, C_23_H_28_ClN_3_O_5_S, molar mass: 493.14 g/mol) [2]. It is an oral anti-hyperglycemic agent, which belongs to a second-generation sulfonylurea group and is used in the treatment of type-II diabetes mellitus by increasing the amount of insulin production from the pancreatic beta cells [2]. It is a weak organic acid due to the acidic NH between the electron-withdrawing groups of the sulfonyl urea functional group (-SO2-NH-CO-N-), and is a biopharmaceutics classification system class II drug with high permeability and low aqueous solubility (0.018 mg·mL^−1^ at 37 °C). It exists as a monoclinic crystalline powder with pK_a_ = 5.3 (at *T* = 25 °C), log *p* = 3.75, hydrogen bond donors (HBD = 3), and hydrogen bond acceptors (HBA = 5) [3]. The two polymorphic forms (solubility: Form I < Form II) have been discovered, where the highly soluble form transforms into the lower soluble form during the manufacturing processes (freeze-drying and high-shear wet granulation), and at certain gastrointestinal (GI) physiological conditions (pH 7.4 to 7.8). Consequently, it affects the dissolution profile, and causes hypoglycemia [4]. Previously, studies were focused on the dissolution improvements of GLN Form I by using various strategies, i.e., milling, polymorphic forms, eutectic mixtures, cocrystals, and β-cyclodextrin complexation while studies on the other aspects i.e., screening, characterization, and stability of GLN Form I during the manufacturing process and physiological conditions were quite limited [5,6,7,8,9]. One of the studies attempted to investigate GLN solvates (GLN pentanol solvate, GLN toluene solvate), but the impact of manufacturing processes i.e., high-shear wet granulation, drying, tableting, and stability under the physiological pH and temperature were not discussed [10]. Although 1,4-dioxane is most widely used solvent, GLN solvate form using 1,4-dioxane has not been reported earlier.

For this study, preliminary solubility data in aqueous and organic solvents is necessary [11]. Such solubility data are crucial for developing oral tablet dosage form, which could provide desired bioavailability [12]. Another study reported its solubility in various solvents, i.e., methanol, ethanol, 1-propanol, 2-propanol, 1-butanol, 2-butanol, acetone, acetonitrile, methyl isobutyl ketone, methyl acetate, and ethyl acetate [2]. Among the studied solvents, methanol, acetonitrile, dimethyl acetamide (DMA), dimethyl formamide (DMF), N-methyl-2-pyrrolidone (NMP), dimethyl sulfoxide (DMSO), and 1,4-dioxane belong to class 2 solvents, while ethanol, 1-propanol, 2-propanol, 1-butanol, 2-butanol, acetone, and ethyl acetate belong to class 3 solvents [13]. Generally, a solvent mixture could contribute to the preferential solvation of the solute [14,15]. Different solvent mixtures such as methanol, ethanol, polyethylene glycol (PEG) 400, acetone, DMF, ethyl acetate, DMA, NMP, and DMSO have all been used to enhance the drug solubility [16,17]. One of the studies reported GLN solubility in NMP + water, propylene glycol + water, and PEG 400 + water mixtures [18,19]. The modified Apelblat model (AM), ideal model, λh model, CNIBS/R-K model, and modified Jouyban–Acree model (JA) have been used to correlate the experimental mole fraction solubility in various organic solvents [20]. The apparent thermodynamic properties of the drug molecule that include the Gibbs free energy change ($\Delta G_{\mathrm{sol}}^{{}^{\circ}}$), enthalpy change ($\Delta H_{\mathrm{sol}}^{{}^{\circ}}$), and entropy change ($\Delta S_{\mathrm{sol}}^{{}^{\circ}}$) are calculated from the solubility data for both organic solvents and solvent mixtures [21,22].

The present study is focused mainly on the screening of GLN solvate forms, which could provide an improved in vitro release profile along with high purity in the tablets with respect to impurities and degradation products during the manufacturing processes (high-shear wet granulation, drying, and tableting) and physiological conditions (pH and temperature) [23,24]. The goal of the study is to determine the solubility of GLN in various organic solvents and aqueous mixtures. GLN solubility data is used to determine the optimum recrystallization conditions and thereby, obtain stable solvate forms [23]. It is not practical to use a too low or too high solubility for recrystallization as a high solubility inhibits crystallization and produces a viscous product, which may be overcome by antisolvent crystallization [25]. Conversely, poor solubility impedes recrystallization, and it may need an amorphous starting material to enhance the solubility. Consequently, a solubility study saves time and resource. Moreover, the slurry technique is possible and is not limited by the drug compound solubility. The GLN solvate recrystallization is conducted on the selected solvents, keeping the preparation methods constant. The obtained solvates from DMSO, NMP, 1,4-dioxane, and Transcutol HP^®^ (THP) were referred to as GLN-DMSOte, GLN-NMPate, GLN-dioxanate, and GLN-THPate hereafter, respectively. A suitable formulation for the solvate form was developed and compressed into tablets and their in vitro release profiles are evaluated. GLN (Form I) is taken as a reference and compressed into tablets using the same formulation ingredients and processing conditions. GLN reference and solvate forms are characterized using differential scanning calorimetry (DSC) thermograms, Fourier transform Infrared spectroscopy (FTIR) spectra, powder X-ray diffraction (PXRD) patterns, scanning electron microscopy (SEM), and in vitro release testing.

## 2. Results and Discussion

### 2.1. Equilibrium Solubility

#### 2.1.1. Solubility in Organic Solvents

The experimental mole fraction solubility (*X*_exp_) of GLN in organic solvents over a range of *T* = (293.15–323.15) K is presented in Figure 2 and Table 1. The solubility increased with the rise in temperature from 293.15 K to 323.15 K, as illustrated in Table 1 (*p* < 0.05). Within the studied temperature range, the order of GLN solubility followed the order DMSO > NMP > 1,4-dioxane > PEG 400 > THP > water (Figure 2). The solubility of GLN in DMSO and NMP was higher compared to 1,4-dioxane, PEG 400, THP, and it dissolved until it was in a colloidal state, which made it difficult to filter and dry the sample. The GLN solubility in methyl isobutyl ketone, 1-butanol, methyl acetate, methanol, ethanol, 1-propanol, ethyl acetate, 2-propanol, acetonitrile, NMP, and PEG 400 had been reported, but GLN solubility in DMSO, 1,4-dioxane, and THP had not been studied earlier [2,18]. Their solubility results were small and of negligible significance for the crystallization process. The solubility in 1,4-dioxane, PEG 400, and THP appeared to indicate adequate solubility for crystallization experiments [2,18]. DMSO, NMP, 1,4-dioxane, PEG 400, and THP showed comparably higher solubility than in water (Figure 2). This could be because of the low dielectric constant or low polarity compared to the other solvents [26]; however, polarity and dielectric constant are not the only factors responsible for increasing the solubility. Dissolution is a complex phenomenon that can be influenced by many factors, including temperature, molecular structure and solvent, molecular size, solvent–solvent interactions, solute–solvent interactions, co-solvent ratio, and the ability to form hydrogen bonds [23,27].

To understand the solvent effect on the solubility, a Kamlet-Taft linear solvation energy relationship (KAT-LSER) model with solvatochromatic parameters (*α*-hydrogen bond donor acidity, *β*-hydrogen bond acceptor basicity, *π**-dipolarity or polarizability), and the Hildebrand solubility parameter (*δ*_H_) were applied as illustrated in Equation (1) [28]. DMSO, NMP, 1,4-dioxane, PEG 400, and THP appeared to be statistically significant (*p* < 0.05) while water appeared to be statistically insignificant (*p* > 0.05). The solvatochromatic parameters for 1,4-dioxane are not adequately reported in the literature, while its solubility in water was much lower among the studied solvents. Hence, only the solvents with statistically significant (*p* < 0.05) findings were reported.
(1)$$\ln\left(X_{\exp}\right)=c_{0}+c_{1}\alpha +c_{2}\beta +c_{3}\pi^{*}+c_{4}\left(\frac{V_{s}\delta_{H}^{2}}{100RT}\right)$$
where *c*_0_ is a constant, *c*_1_ and *c*_2_ are the susceptibility of the solute to solute–solvent interactions via hydrogen bonding, *c*_3_ and *c*_4_ are the susceptibility of the solute to electrostatic solute–solvent and solvent–solvent interactions, and *R*, *T*, and *V_s_* are the universal gas constant (8.314 J⋅K^−1^⋅mol^−1^), absolute temperature, and molar volume of solute respectively. The *Vs* value for GLN was calculated as 28.38 MPa^1/2^ based on Fedors’ method, as illustrated in Table 2. The parameters *α*, *β*, *π**, and *δ*_H_ were taken from published articles and listed in Table 3 [28,29]. The KAT-LSER model coefficient values with their standard error were estimated from a multiple linear regression analysis of the experimental and ideal mole fraction solubility data at 298.15 K, as illustrated in Equation (2).
(2)$$\ln\left(X_{\exp}\right)=-6.17\left(3.35\right)-6.75\left(1.55\right)\alpha +11.51\left(3.86\right)\beta +\;7.15\left(3.01\right)\pi^{*}-1.22\left(1.83\right)\left(\frac{V_{s}\delta_{H}^{2}}{100RT}\right)$$

Based on the estimated coefficients, the parameters *α*, *β*, *π**, and *δ*_H_ were 20.58%, 35.09%, 21.80%, and 3.72%, respectively. The *β* and *π** were positive, while *α* and $\left(\frac{V_{s}\delta_{H}^{2}}{100RT}\right)$ were negative, which suggested that the interactions of the solvent with the solute (decreased hydrogen bonding acidity and increased hydrogen bond basicity), increased electrostatic solute–solvent interactions, and decreased solvent–solvent interactions contributed to higher GLN solubility. The solute–solvent interactions are predominantly hydrogen bond basicity and the electrostatic solute–solvent interactions appeared to contribute more than the solvent–solvent interactions.

Experimental solubility data in each solvent were evaluated using a modified AM model, ideal model; *λh* model and the parameters along with the *RMSD* and *MRD* values are listed in Table 4. The smaller *RMSD* (4.347) and *MRD* (0.124) values in the modified AM model indicated good agreement between the calculated and the experimental solubility among the studied models (Table 4). The AM showed a smaller *RMSD* (0.078) except for NMP *RMSD* (0.032), and a smaller *MRD* (0.164) except for DMSO *MRD* (0.001) than the other studied solvents.

#### 2.1.2. Solubility in Solvent Mixtures and Solid State Stability

The values of the mole fraction solubility of GLN in binary mixtures, namely (DMSO + water), (NMP + water), (1,4-dioxane + water), (PEG 400 + water), and (THP + water) mixtures at 298.15 K, are provided in Figure 3. The mole fraction solubility appeared to increase with a higher mole fraction of 1,4-dioxane in the mixture at 318.15 K. It suggested the highest solubility at 0.8 mole fraction (1.93 × 10^−1^), which decreased gradually at 1.0 mole fraction of 1,4-dioxane (1.35 × 10^−1^), indicating preferential solvation at that ratio. A Perturbed-Chain Statistical Associating Fluid Theory (PC-SAFT) study previously confirmed that screening of a crystalline drug compound in the solvent mixture could discover respective soluble ones, which remained stable and did not undergo phase separation [30]. The preferential solvation was not observed with DMSO, NMP, PEG 400, and THP [17]. The characteristics of a higher solubility at a certain mole fraction of 1,4-dioxane + water mixture was further evaluated in a temperature range of 293.15 K to 323.15 K, as shown in Table 5 which suggested the solubility was maximum at 0.8 mole fraction of 1,4-dioxane in water. The experimental mole fraction solubility was optimized against a temperature range of 293.15 K to 323.15 K and mole fraction of 1,4-dioxane in water as shown in Figure 4, which demonstrated the case of preferential solvation at 1,4-dioxane (*w* = 0.8) in the 1,4-dioxane + water mixture. The GLN solubility increased proportionately as the temperature increased from 293.15 K to 323.15 K (Figure 4). The solid state stability of the formed solvate remained unchanged up to 323.15 K. This resulted in a 1,4-dioxane aqueous mixture that has not been previously reported.

In the cases of DMSO, NMP, PEG 400, and THP, the mole fraction solubility increased as the mole fraction of the respective solvent in the binary mixture increased. In DMSO and NMP, the mole fraction solubility increased rapidly at 0~0.5 and 0.7~1.0 and slowly in the 0.5~0.7 mole fraction (Figure 3). This may indicate the importance of co-solvency to improve the solubility of GLN [18]. Furthermore, the solubility of a solute in a mixed solvent is influenced by several factors, such as polarity, temperature, mole fraction of solutes, and solvents [31]. Previously, the milled GLN, a eutectic mixture with L-arginine, and the particular polymorphic form to improve GLN’s solubility and in vitro release profile were studied [4,5,7]. In the present study, GLN solubility was investigated and the feasibility to recrystallize into solvate form to improve its release profile and stability was discussed.

The solubility of GLN in 1,4-dioxane + water mixture was evaluated using a modified AM model, ideal model (Table 6), CNIBS/R-K model, and JA model (Table 7). The parameters were evaluated and the deviations of modified AM (*RMSD* 0.519, *MRD* 0.664), ideal model (*RMSD* 0.675, *MRD* 2.218), CNIBS/R-K model (*RMSD* 0.358, *MRD* 3.936), and JA model (*RMSD* 0.689, *MRD* 4.247) were obtained, of which CNIBS/R-K model had smaller deviation and indicated a good agreement between the experimental and calculated data of GLN solubility in the 1,4-dioxane + water mixture (Table 6, Table 7). The modified AM model only considered the temperature, not the mole fraction of the co-solvent; therefore, the CNIBS/R-K model was considered optimal for the 1,4-dioxane + water mixture.

### 2.2. Ideal Solubility and Activity Coefficient

The activity coefficients (*γ*_i_) were calculated using Equation (17) to study the molecular interactions between GLN and the respective organic solvent and are illustrated in Table 8. The *γ*_i_ values decreased significantly with the rise in temperature (*p* < 0.05). The *X*^ID^ values were found to be significantly lower than the *X*_exp_ in DMSO, NMP, and 1,4-dioxane, while the *X*^ID^ values were significantly higher than the *X*_exp_ in PEG 400, THP, and water (Table 8). At higher temperatures, the *X*^ID^ value in 1,4-dioxane was close to the *X*_exp_ of GLN while DMSO and NMP gave significantly lower *X*^ID^ values than *X*_exp_ values. The *X*^ID^ values of GLN in PEG 400, THP, and water at higher temperatures were significantly higher than the *X*_exp_ values of GLN. The *γ*_i_ values in 1,4-dioxane were near to unit in a range of *T* = (293.15~323.15) K, which suggested an ideal behavior. Based on these data (Table 8), it could be suggested that the higher solute–solvent interactions at molecular level were found and GLN-1,4-dioxane interaction prediction was near to experimental values. The mole fraction solubility was further studied in 1,4-dioxane + water in *T* = (293.15~323.15) K, as illustrated in Figure 4.

### 2.3. Apparent Thermodynamic Analysis

A thermodynamic analysis was performed to evaluate the dissolution behavior of GLN in various organic solvents and 1,4-dioxane + water mixture [32]. The $\Delta H_{\mathrm{sol}}^{{}^{\circ}}$, $\Delta G_{\mathrm{sol}}^{{}^{\circ}}$, and $\Delta S_{\mathrm{sol}}^{{}^{\circ}}$ of GLN in solution were obtained with Equations (3)–(5) [33]:(3)$$\Delta H_{\mathrm{sol}}^{{}^{\circ}}=-R\left(\frac{\partial \ln x_{\exp}}{\partial \left(1/T-1/T_{\mathrm{hm}}\right)}\right)$$
where $X_{\exp}$ is the mole fraction solubility, R is the universal gas constant (8.314 J·mol^−1^·K^−1^); $T_{\mathrm{hm}}$ is the mean harmonic temperature from (293.15~323.15) K, and the value is 308.15 K. The logarithmic mole fraction solubility of the GLN ($\ln X_{\exp}$) is linearly related to the reciprocal of the absolute temperature (1/*T*). The slope of the plot of $\ln x_{\exp}$ against $\left(1/T-1/T_{\mathrm{hm}}\right)$ gives the value of (−*ΔH*°_sol_/*T*) and the intercept helps in the calculation of $\;\Delta G_{\mathrm{sol}}^{{}^{\circ}}$ as expressed in Equation (4):(4)$$\Delta G_{\mathrm{sol}}^{{}^{\circ}}=-RT_{\mathrm{hm}}\;\times \mathrm{intercept}$$

Finally, the entropy change ($\Delta S_{\mathrm{sol}}^{{}^{\circ}})\;$ of drug dissolution can be obtained using Equation (5):(5)$$\Delta S_{\mathrm{sol}}^{{}^{\circ}}=\left(\frac{\Delta H_{\mathrm{sol}}^{{}^{\circ}}-\Delta G_{\mathrm{sol}}^{{}^{\circ}}\;}{T_{\mathrm{hm}}}\right)$$DMSO, NMP, 1,4-dioxane, PEG 400, and THP gave negative values of $\Delta H_{\mathrm{sol}}^{{}^{\circ}}$ suggesting exothermic process ($\Delta H_{\mathrm{sol}}^{{}^{\circ}}$ < 0) whereas water gave positive values of $\Delta H_{\mathrm{sol}}^{{}^{\circ}}$ suggesting endothermic process ($\Delta H_{\mathrm{sol}}^{{}^{\circ}}$ > 0) (Table 9). The GLN dissolvability increased with the rise in temperature. High values of $\Delta H_{\mathrm{sol}}^{{}^{\circ}}$ reflected the strong temperature-dependent solubility and Table 9 showed that GLN solubility in water ($\Delta H_{\mathrm{sol}}^{{}^{\circ}}$ = 30.33) is strongly dependent on the temperature [34]. Moreover, positive $\Delta H_{\mathrm{sol}}^{{}^{\circ}}$ indicated that the molecular interactions between GLN and the solvents was stronger and required higher energies for breaking solute-solute and solvent–solvent intermolecular interactions [35]. Similarly, the decreased value of $\Delta G_{\mathrm{sol}}^{{}^{\circ}}$ indicates that the dissolution process is more favorable in the solvents with high solubility [16]. It was found that the $\Delta G_{\mathrm{sol}}^{{}^{\circ}}$ values were negative in DMSO and NMP ($\Delta G_{\mathrm{sol}}^{{}^{\circ}}$ < 0), suggesting spontaneous process while the $\Delta G_{\mathrm{sol}}^{{}^{\circ}}$ values were positive in 1,4-dioxane, PEG 400, THP, and water ($\Delta G_{\mathrm{sol}}^{{}^{\circ}}$ > 0), suggesting non-spontaneous process (Table 9). The negative $\Delta S_{\mathrm{sol}}^{{}^{\circ}}$ values ($\Delta S_{\mathrm{sol}}^{{}^{\circ}}$ < 0) obtained with DMSO, NMP, 1,4-dioxane, PEG 400, and THP suggested an enthalpy-driven process whereas the positive $\Delta S_{\mathrm{sol}}^{{}^{\circ}}$ values ($\Delta S_{\mathrm{sol}}^{{}^{\circ}}$ > 0) obtained in water suggested entropy-driven process (Table 9) [21]. In general, Table 9 showed that GLN solubility in 1,4-dioxane was exothermic ($\Delta H_{\mathrm{sol}}^{{}^{\circ}}$ = −9.65), non-spontaneous ($\Delta G_{\mathrm{sol}}^{{}^{\circ}}$ = 4.42), and enthalpy-driven ($\Delta S_{\mathrm{sol}}^{{}^{\circ}}$ = −45.70) whereas GLN solubility in water was endothermic ($\Delta H_{\mathrm{sol}}^{{}^{\circ}}$ = 30.33), non-spontaneous ($\Delta G_{\mathrm{sol}}^{{}^{\circ}}$ = 28.96), and entropy-driven ($\Delta S_{\mathrm{sol}}^{{}^{\circ}}$ = 4.44) process.

The solvation behavior in various 1,4-dioxane + water mixture was evaluated using enthalpy–entropy compensation analysis, as illustrated in Figure 5. It was found that a positive slope was observed in the interval 0 < $x_{2}$ < 0.10, 0.20 < $x_{2}$ < 0.70 and 0.80 < $x_{2}$ < 1.00, whereas a negative slope was observed in the interval 0.10 < $x_{2}$ < 0.20 and 0.70 < $x_{2}$ < 0.80. This might be because of the maximum solvation in the 1,4-dioxane-rich mixture [36,37]. Similarly, Table 10 showed that enthalpy ($\Delta H_{\mathrm{sol}}^{{}^{\circ}}$ = 52.89 → 7.26 kJ·mol^−1^), Gibb’s free energy ($\Delta G_{\mathrm{sol}}^{{}^{\circ}}$ = 5.75 → −19.63 kJ·mol^−1^), and entropy ($\Delta S_{\mathrm{sol}}^{{}^{\circ}}$ = 153.08 → 87.32 J·mol^−1^·K^−1^) decreased as mole fraction of 1,4-dioxane (*w*) was gradually increased in the (1,4-dioxane + water) mixture up to 0.8 mole fraction of 1,4-dioxane. After the 0.8 mole fraction of 1,4-dioxane, all 3 parameters ($\Delta H_{\mathrm{sol}}^{{}^{\circ}}$, $\Delta G_{\mathrm{sol}}^{{}^{\circ}}$, $\Delta S_{\mathrm{sol}}^{{}^{\circ}}$) started to increase. The lower thermodynamic parameters ($\Delta H_{\mathrm{sol}}^{{}^{\circ}}$, $\Delta G_{\mathrm{sol}}^{{}^{\circ}}$, $\Delta S_{\mathrm{sol}}^{{}^{\circ}}$) suggested improved solubility, as can be seen in Figure 3 and Figure 4.

### 2.4. Characterization of Solvate

The DSC thermogram of GLN reference is shown in Figure 6a(i). The melting temperature ($T_{\mathrm{fus}}$, 446.42 ± 0.26 K) and the enthalpy of fusion ($\Delta H_{\mathrm{fus}}$, 50.94 ± 0.57 kJ·mol^−1^) were obtained and comparison with the reported values (445.2~447.2 K) confirmed GLN crystals (Form I) [10,18]. The recovered solid solvate crystals from the bottom of each saturated solution of GLN-dioxanate, GLN-DMSOte, GLN-NMPate, and GLN-THPate gave endothermic peaks at 443.16 K, 473.74 K, 469.53 K, and 380.76 K and enthalpies of fusion as 18.33, 31.92, 47.61, and 69.34 kJ·mol^−1^, respectively (Figure 6a(ii)–(v)). The thermal properties of the recovered solvate crystals were significantly different compared to that of the GLN reference (*p* < 0.05) while solvate crystals from PEG 400 could not be obtained from the present procedure. GLN-dioxanate had lower melting point (443.16 K) and higher solubility (Table 1) than the GLN Form I (reference), indicating the attributes of hydrogen bonding [8]. It needs to be considered that negative experimental result of a solvate screening may not necessarily exclude the possibility of certain solid form. There could be various reasons for not observing in the specific experimental setup. The presence of minor impurities, lack of adequate solubility due to solute–solvent interactions, or the existence of highly stable hydrate (masking the existence of solvate) may inhibit nucleation of the solvate form [1,38].

Previously, it was confirmed that heating GLN alone and then quickly cooling would give a thermal degradation product, i.e., 1,3-dicyclohexylurea, and it did not represent a GLN solvate [39,40]. On the contrary, crystalline GLN solids heating along with the solvent (pentanol, toluene) close to its boiling point produced GLN Form I pentanol solvate, and GLN Form I toluene solvate [10]. In this study, the GLN reference was heated close to the boiling point of the respective solvents. When it was heated close to the boiling point of DMSO or NMP, it gave a product with a melting point at 473.74 K and 469.53 K, respectively, which were close to Form I decomposition peak (489.15 K) suggesting DMSO and NMP solvates may not be in pure form [4]. In addition, the GLN-THPate crystals had quality issues with respect to handling and processing. In conclusion, the obtained GLN-dioxanate solvate suggested to be in a pure form (Form I) [41].

The FTIR spectra of GLN showed characteristic peaks at 1684 cm^−1^ (C-C stretching), 3060 cm^−1^ (C-H stretching), 1680 cm^−1^ (O=C-NH), 1550 cm^−1^ (N-H), 2900 cm^−1^ (C_6_H_12_), 1600 cm^−1^ (C=O) 1450 cm^−1^ (C=C stretching), 746 cm^−1^ (C-Cl stretching) as compared against GLN reference (Figure 6b(i)) and the recovered GLN solvate crystals (DMSOte, NMPate, dioxanate, and THPate) (Figure 6b(ii)–(v)) [39]. The main difference in the FTIR spectra of the solvates was in the region from 3000 to 3500 cm^−1^ where Form I showed 2 absorption bands at 3290 and 3370 cm^−1^ whereas Form II showed only one band at 3330 cm^−1^ and an additional wide band at 3100 cm^−1^ [41]. The FTIR curve corresponded to Form I (thermodynamically stable form) in both the GLN reference and solvate form, which further supported our DSC results [4].

The PXRD pattern of GLN reference (Figure 6c(i)) presented characteristic crystalline peaks at 8.240°, 7.295°, 5.334°, 4.592°, 4.479°, 4.161°, and 3.779° [4]. The characteristic diffraction peaks of GLN were observed in the recovered solid solvate crystals (dioxanate, DMSOte, NMPate, and THPate) (Figure 6c(ii)–(v)). The obtained solvates were tested under fluctuating conditions (298.15 K and 323.15 K), and the corresponding PXRD readings were unchanged (data not shown) [39]. The SEM image of the reference, as shown in Figure 6d(i) was similar to the crystalline solids (Form I) [39]. The SEM images of the solvate crystals (dioxanate—Figure 6d(ii), NMPate—Figure 6d(iii), DMSOte—Figure 6d(iv), and THPate—Figure 6d(v)) suggested the formation of their respective solvate crystals. The obtained results from the DSC thermograms, FTIR spectra, PXRD pattern, and SEM images of the reference and solvates (dioxanate, NMPate, DMSOte, and THPate) corroborated the obtained polymorphic crystalline Form I [4,8,18].

### 2.5. Tableting of GLN Solvate, In Vitro Dissolution Study, and Stability

The in vitro drug release amounts of the GLN reference, dioxanate, DMSOte, NMPate, and THPate at 0 day were 28.93 ± 4.32%, 95.51 ± 1.31%, 87.35 ± 2.26%, 85.56 ± 2.21%, and 83.01 ± 1.21%, respectively (Figure 7). The samples were kept at 40 °C (313.15 K) 75% RH for 30-day and the in vitro drug release amounts of the GLN reference, dioxanate, DMSOte, NMPate, and THPate were obtained as 24.87 ± 4.29%, 93.74 ± 2.61%, 75.33 ± 2.34%, 73.65 ± 3.46%, and 80.37 ± 1.48%, respectively (Figure 7). In conclusion, the dioxanate and THPate seem to be stable up to 30-day. On the contrary, the dissolution of DMSOte and NMPate significantly decreased by 12.02% and 11.91%, respectively. The probable cause of reduced dissolution in DMSOte and NMPate might be due to the impurities formed upon degradation (DMSO boiling point 462.15 K, NMP boiling point 475.15 K) compared to dioxanate and THPate, as those GLN solvates decomposed at 473.74 K and 469.53 K respectively which were close to the Form I decomposition peak (489.15 K) [4]. Even a small amount of impurity in the slurry could impede polymorphic stability [42]; however, the degradation of the solvates kept at 40 °C and 75% RH for 30-day, had less than 1.5% impurity, indicating a high degree of purity of the GLN solids with respect to impurities and degradation products [43]. Previously, GLN binary mixture gave more hydrophilic structure of the crystal surface compared to GLN reference (dissolution of *X*_G_ = 0.3, 0.5 or 0.7 > dissolution of *X*_G_ = 1) which increased its wettability and contributed to improved dissolution [7]. In our case, GLN-dioxanate binary mixture suggested more hydrophilic in nature as it gave higher release profile among the solvates studied [7,44]. High-shear wet granulation and freeze-drying methods did not affect the stability of the GLN-dioxanate in the present experiment. In large-scale production, the granulation and drying methods are guided with good manufacturing practice in the pharmaceutical industry and thus, quality and stability could be expected [45,46].

Solvent-mediated slow crystallization under thermodynamic conditions could produce the most stable form of the compound [42,47]. Although GLN Form II was observed to be about 3.5 times more soluble (in vitro data) and released 2 times (in vivo data) more drug than the Form I in the physiological pH range, the polymorphic transformation from Form II (metastable) to Form I was reported in pH range of 7.4 to 7.8 [4]. Another study reported that the in vitro release of GLN was different when the amount of sodium lauryl sulfate (surfactant) was varied in the phosphate buffer pH 6.8 [6]. In the present study, the GLN-dioxanate tablets gave higher cumulative release profile (Figure 7) among the studied solvates, equivalent to Form II release profile, without addition of the surfactant [6]. Moreover, in vitro release profile obtained after 30-day suggested that it was stable and not affected by high-shear wet granulation, freeze-drying, or at physiological pH. In conclusion, the GLN-dioxanate solvate form could be a suitable alternative for improving the molecule’s properties.

## 3. Experimental Section

### 3.1. Materials

GLN was obtained from Sigma Aldrich, Inc. (St. Louis, MO, USA; purity > 0.998 in mass fraction). Its purity was supported by its melting point (446.42 K). DMSO (purity of at least 0.999 in mass fraction), NMP (purity of at least 0.995 in mass fraction), 1,4-dioxane (purity of at least 0.998 in mass fraction), and PEG 400 (purity of at least 0.995 in mass fraction) were purchased from Daejung Chemical & Metals Co., Ltd. (Siheung, Gyeonggi, Korea). THP was obtained from Gattefosse (Cedex, France). Kollidon^®^ CL was obtained from BASF (Ludwigshafen, Germany). Avicel^®^ PH-102 was obtained from Sigma Aldrich (Darmstadt, Germany), Pharmatose^®^ 130M DFE Pharma (Klever Strasse, Germany). Povidone K30 and sodium starch glycolate were obtained from JRS Pharma GmbH (Rosenberg, Germany). Magnesium stearate was obtained from Faci Asia Pacific Pte. Ltd. (Jurong Island, Singapore). Water was from a Milli-Q water purifier (Millipore, Lyon, France). The reagents were of analytical or high-performance liquid chromatography (HPLC) grade and were used without further purification.

### 3.2. High-Performance Liquid Chromatography (HPLC)

The GLN reference and obtained solvate products (crystal and tablet dosage form) were quantified using an HPLC system (LC-20AD, Shimadzu, Kyoto, Japan) with an Eclipse plus C_18_ column (3.9 mm × 150 mm, 5 µm) set at a *T* = 303.15 K and an ultraviolet (UV) detector at 230 nm. The mobile phase was a mixture of 10 mM phosphate buffer pH 2.6 and acetonitrile (50:50, *v*/*v*). The flow rate of the mobile phase was 1.0 mL·min^−1^, and the injection volume was 20 µL [4]. All measurements were performed in triplicate. The standard calibration curve was found linear in the range of 0.1 to 4.0 μg·mL^−1^ with a correlation coefficient of 0.9999.

### 3.3. GLN Solubility Determination with Experimental Approach

The solubility of GLN, Form I was studied in various solvents (1,4-dioxane, NMP, DMSO, THP, PEG 400, and water) and in their respective aqueous binary mixture using the static equilibrium method at *T* = (293.15~323.15) K [36]. Briefly, an excess amount of solid GLN was added to the known amount of solvent with an uncertainty of 0.001 g. The same procedure was used to measure its solubility in a binary mixture at various temperatures. The solute–solvent mixture was vortexed for 10 min using a vortex shaker (Daihan Scientific, Seoul, Korea). It was placed in a shaking water bath (Jeiotech Co., Ltd., Daejeon, Korea) at 100 rpm for 72 h to reach equilibrium. The equilibrium time was optimized based on preliminary studies. The water bath was provided with a thermostat (Shanghai Laboratory Instrument Works, Shanghai, China) capable of maintaining the temperature within ±0.05 °C. At the end of the experiment, the samples were removed from the shaker and left to sit for about 98 h to allow the undissolved particles to settle down [18]. It was then centrifuged at 10,000 rpm for 15 min (Eppendorf Inc., Westbury, CT, USA). The supernatants were then filtered through a 0.45 μm polytetrafluoroethylene (PTFE) syringe filter (Hyundai Micro, Seoul, Korea) and appropriately diluted with methanol, before analysis. The amount of drug in each sample was obtained from the standard plot of GLN.

Each experimental data point represented the arithmetic average of at least three repetitive experiments. The density of the saturated solution was measured using a 5 mL pycnometer. It was necessary to convert the molar solubility to the mole fraction solubility. The experimental mole fraction solubility ($X_{\exp}$) of GLN in the organic solvents was calculated using Equation (6) [12]:(6)$$X_{\exp}=\frac{m_{A}/M_{A}}{m_{A}/M_{A}+m_{1}/M_{1}}$$
where $m_{A}$ and $m_{1}$ are the mass of the GLN and solvent, and $M_{A}$ and $M_{1}$ are the respective molar masses of GLN and the solvent, respectively.

The mole fraction of solvent (*w*) in the aqueous binary mixture varied from 0.1 to 0.9 and it could be obtained using Equation (7) [12]:(7)$$w=\frac{m_{2}}{m_{2}+m_{1}}$$
where $m_{1}$ and $m_{2}$ represent the masses of the water and solvent, respectively. Similarly, the mole fraction solubility of GLN ($X_{\exp}$) in the binary mixture of the water and solvent at different temperatures can be obtained by Equation (8) [12]:(8)$$x_{\exp}=\frac{m_{A}/M_{A}}{m_{A}/M_{A}+m_{1}/M_{1}+m_{2}/M_{2}}$$
where $m_{A},$ $m_{1}$, and $m_{2}$ are the masses of the GLN, water, and solvent; $M_{A}$, $M_{1}$, and $M_{2}$ are the molar masses of the GLN, water, and solvent. The experiment was carried out in triplicate and the arithmetic average was used as the final value.

### 3.4. Computational Validation of the Experimental Data

The solubility of GLN in an organic solvent and its aqueous binary mixture were analyzed and correlated using a modified AM model, Ideal model, *λh* model, CNIBS/R-K model, and modified JA model.

#### 3.4.1. Modified Apelblat Model (AM)

Modified AM is a semi-empirical model, in which Equation (9) correlates the mole fraction solubility and the absolute temperature for both the polar and non-polar solvents. It can be expressed as [48,49]:(9)$$\ln X_{\mathrm{AM}}=A+\frac{B}{T}+C\;\ln\left(T\right)\;$$
where $X_{\mathrm{AM}}$ is the mole fraction solubility at *T*/K, and *A*, *B*, and *C* are the model parameters obtained by non-linear regression equation, where the parameters *A* and *B* represent the non-ideal behavior of the solution in terms of the activity coefficient variation in the solution, and *C* reflects the temperature effect on the enthalpy of fusion.

#### 3.4.2. Ideal Model

In the equation illustrated in Equation (10), the logarithm of the mole fraction solubility was linearly correlated to the reciprocal of the absolute temperature in the ideal solution. It was a simplified expression of the activity coefficient formula and is expressed as [22]:(10)$$\ln X_{\mathrm{IDL}}=A+\frac{B}{T}$$
where *T* is the absolute temperature, $X_{\mathrm{IDL}}$ is the mole fraction solubility, and *A* and *B* are the model parameters.

#### 3.4.3. λh Model

To describe the solid-liquid equilibrium behavior, the *λh* equation was developed and Equation (11) was obtained by Buchowski. The equation is expressed as [36]:(11)$$\ln\left[1+\frac{\lambda \left(1-X_{\lambda h}\right)}{X_{\lambda h}}\right]=\lambda h\left[\frac{1}{T}-\frac{1}{T_{m}}\right]$$
where $X_{\lambda h}$ is the mole fraction solubility, T is the experimental absolute temperature, and $T_{m}$ is the melting temperature in Kelvin (K). The value of $T_{m}$ was found to be 446.42 K in the thermal analysis. The λ and h are the model parameters.

#### 3.4.4. CNIBS/R-K Model

In the CNIBS/R-K model illustrated in Equation (12), the logarithm of the mole fraction solubility was linearly correlated with the solvent composition, in the 1,4-dioxane and water mixture. The equation was expressed as [50,51]:(12)$$\ln X_{\mathrm{RK}}=x_{2}^{0}\ln{\left(x_{1}\right)}_{2}+x_{3}^{0}\ln{\left(X_{\mathrm{RK}}\right)}_{3}+x_{2}^{0}x_{3}^{0}\sum_{i=0}^{N}S_{i}{\left(x_{2}^{0}-x_{3}^{0}\right)}^{i}\;$$
where $X_{\mathrm{RK}}$ is the mole fraction solubility and $x_{2}^{0}$ and $x_{3}^{0}$ are the initial mole fraction composition of 1,4-dioxane and water in the solvent mixture before adding GLN. $S_{i}$ is the model parameter. *N* is the number for the curve-fit coefficient. For the binary mixture, *N* = 2 and $x_{3}^{0}=1-x_{2}^{0}$; thus, the CNIBS/N-K model can be expressed in Equation (13):(13)$$\ln X_{\mathrm{RK}}=B_{0}+B_{1}x_{2}^{0}+B_{2}{\left(x_{2}^{0}\right)}^{2}+B_{3}{\left(x_{2}^{0}\right)}^{3}+B_{4}{\left(x_{2}^{0}\right)}^{4}$$
where $B_{0}$, $B_{1}$, $B_{2}$, $B_{3}$, and $B_{4}$ are the model constants obtained by least-squares regression.

#### 3.4.5. Modified Jouyban–Acree Model

The modified JA model related the solubility to both the temperature and solvent composition. The simplified model was expressed as follows [2]:(14)$$\ln X_{\mathrm{JA}}=A_{1}+\frac{A_{2}}{T}+A_{3}\ln T+A_{4}x_{2}+A_{5}\frac{x_{2}}{T}+A_{6}\frac{x_{2}^{2}}{T}+A_{7}\frac{x_{2}^{3}}{T}+A_{8}\frac{x_{2}^{4}}{T}+A_{9}x_{2}\ln T$$
where, A_1_ to A_9_ are model parameters of the modified Jouyban–Acree model. The T and $x_{2}$ are the absolute temperature and the initial mole fraction composition of 1,4-dioxane in the mixture before GLN addition.

#### 3.4.6. Data Correlation

To distinguish the experimental and calculated solubility data, the mean relative deviation (*MRD*) and relative mean standard deviation (*RMSD*) were used, and expressed as [15,52]:(15)$$MRD\;\left(\%\right)=\frac{100}{N}\;\sum (\frac{\left[X_{\exp}-X_{\mathrm{cal}}\right]}{X_{\exp}})$$
(16)$$RMSD=\sqrt{\frac{\sum_{i=1}^{N}{\left(X_{\exp}-X_{\mathrm{cal}}\right)}^{2}}{N}}$$
where *N* is the number of experimental data points, and $X_{\exp}$ and $X_{\mathrm{cal}}$ represent the experimental value and the calculated value of the mole fraction solubility, respectively.

### 3.5. Ideal Solubility and the Activity Coefficient

The *X*^ID^ value of GLN was calculated with Equation (17) [53,54]:(17)$$\ln X^{\mathrm{ID}}=\frac{-\Delta H_{\mathrm{fus}}\left(T_{\mathrm{fus}}+T\right)}{RT_{\mathrm{fus}}T}+\left(\frac{\Delta C_{p}}{R}\right)\left[\frac{T_{\mathrm{fus}}-T}{T}+\ln\left(\frac{T}{T_{\mathrm{fus}}}\right)\right]$$
where *R* = universal gas constant. 

The ∆*C*p was calculated with Equation (18): (18)$$\Delta C_{p}=\Delta H_{fus}/T_{fus}$$

The $T_{\mathrm{fus}}$ and $\Delta H_{\mathrm{fus}}$ values were calculated as 446.42 K and 50.94 kJ·mol^−1^ respectively. The ∆*C*_p_ was obtained as 113.55 J·mol^−1^ K^−1^. The *X*^ID^ value could then be calculated with Equation (18). The $\gamma_{i}$ value was calculated with Equation (19) [53]:(19)$$\gamma_{i}=X^{\mathrm{ID}}/X_{\exp}$$
where *X*^ID^ and *X*_exp_ are ideal and experimental mole fraction solubility in the respective solvents.

### 3.6. Preparation of the GLN Solvate

The excess Form I solids (usually 1 g) were dissolved in 90 mL of each of the different solvents separately and heated close to the respective boiling point of the solvent (DMSO 462.15 K, NMP 475.15 K, 1,4-dioxane 374.15 K, PEG 400 473.15 K, and THP 475.15 K) [1]. The solution was filtered through a filter paper to get rid of the insoluble particles, and then cooled down slowly to 273.15 K. The obtained product (GLN-dioxanate) from the 1,4-dioxane solvent, was dried in a vacuum oven at 313.15 K and 150 mbar pressure for 24 h [1]. The products from DMSO, NMP, and THP were difficult to isolate using a vacuum oven and were instead obtained using a freeze dryer (Operon, Yangchon, Korea) for 72 h. A product could not be obtained from PEG 400. The resulting crystals were sieved using a stainless-steel mesh (with fractions between a 200 mesh (75 μm) and a 270 mesh (53 μm) to ensure homogeneous particle size), passed under a current of nitrogen gas, and stored in a desiccator over silica for at least 24 h until further use [4,10].

### 3.7. Solid State Stability of the GLN Solvate

The phase transformation can occur in the solvate solid state as a response to temperature variation. The effect of such environmental factor (*T*) on the quality and stability of an obtained solvate is evaluated [25]. Each solvate (DMSO, NMP, 1,4-dioxane, and THP) was transferred into a 1.5 mL glass vial and a suspension was prepared. The amount of excess solid solvate form (usually 10~50 mg) in the 1 mL solvent, could vary depending on the solvate solubility. The suspension was vortexed at 750 rpm, using a shaking water bath (Jeiotech Co., Ltd., Daejeon, Korea) at 298.15 K for 2 weeks [25]. It was centrifuged at 10,000 rpm for 15 min (Eppendorf Inc., Westbury, CT, USA) and the supernatants were filtered through a 0.45 μm PTFE syringe filter (Hyundai Micro, Seoul, Korea). It was appropriately diluted with methanol before analysis. Quantification of the samples was carried out using HPLC method. All measurements were performed in triplicate, where the average values were used to calculate the mole fraction solubility of the GLN.

The identity of the solvate solid phases could be evaluated by their PXRD patterns. Based on ICH Guideline Q3C, the preferred class 3 solvent has a daily exposure limit up to 50 mg. A higher amount could be acceptable if the manufacturer proved that the amount was realistic and in accordance with good manufacturing practices [55]. Furthermore, solid form screening should not be limited to ICH class 3 solvents only. The solvates formed with a class 2 or 3 solvent could give a valuable alternative form [56]. The present study was focused more on the main crystalline solvate of GLN (DMSO, NMP, 1,4-dioxane, and THP).

### 3.8. Characterization of GLN Solvate

The melting temperature and enthalpy of fusion for the reference and the solvate were determined using DSC (TA Instruments, New Castle, DE, USA). A sample (2 mg) was weighed (Mettler Toledo, Greifensee, Switzerland) and sealed in a T_zero_ aluminum pan. A blank pan was employed as a reference. The DSC measurements were carried out at a scan rate of 10 K·min^−1^ in a range of *T* = (293.15~573.15) K under a nitrogen flow of 50 mL·min^−1^. The standard uncertainty of the melting temperature was estimated to be 0.5 K. Various thermal parameters were obtained and interpreted using the software provided with the instrument. 

A thermal analysis was performed to analyze the different thermal parameters and to evaluate the possible GLN transformation into the solvate form [21,32]. FTIR spectra were measured using a FTIR spectrophotometer (Thermo Fisher Scientific, MA, USA). Each sample was analyzed using 32 scan rates at a resolution of 4 cm^−1^ over a wavenumber region of 4000~500 cm^−1^. The PXRD patterns were measured using a D2 phaser benchtop X-ray diffractometer (Bruker AXS GmbH, Karlsruhe, Germany) equipped with Ni-filtered Cu-Kα radiation (λ = 1.54056 Å) and a high speed LynxEye detector. The powdered samples were placed in a quartz holder and scanned over a range of 4~40° at a scanning rate of 6°/min. 

### 3.9. Scanning Electron Microscope

The morphology of the dry sample was examined with a SEM (COXEM, Daejeon, Korea) at an accelerating voltage of 20 kV. The samples were initially coated with gold under vacuum in an argon atmosphere using an ion coater (Hitachi, Tokyo, Japan) before the examination. 

### 3.10. High-Shear Wet Granulation

A small-scale (55 mL) Mi-Pro high-shear granulator (ProCepT, Zelzate, Belgium) was used to prepare the granules of the GLN reference, GLN-DMSOte, GLN-NMPate, GLN-dioxanate, and GLN-THPate. Each sample was mixed with 22% *w*/*w* Kollidon^®^ CL, 30% *w*/*w* Avicel^®^ 102, 45% *w*/*w* Pharmatose^®^ 130M for 3 min, passed through a 42 mesh (355 μm) screen, and granulated with 2% *w*/*w* Povidone K30 solution using a syringe pump, and kneaded with an impeller at 500 rpm. After the granulation, the obtained granules were sieved and dried in a freeze dryer (Ilshin Bio Base, Yangju, Korea) until the constant weight was achieved. The resulting granules were passed under a current of nitrogen gas and stored in a desiccator over silica for at least 24 h. 

### 3.11. Tableting of Solvate, and Its Dissolution and Stability

GLN reference granule, GLN-DMSOte granule, GLN-NMPate granule, GLN-dioxanate granule, and GLN-THPate granule were blended with 2% *w*/*w* sodium starch glycolate, lubricated with 0.5% *w*/*w* magnesium stearate, and compressed into a tablet using a single-punch Carver Laboratory Press (Carver Inc., Wabash, IN, USA) (flat-faced with beveled edge, 6 mm diameter, 100 mg average weight, tableting pressure 100 MPa). The tablet was accurately weighed, and the diameter and out-of-die thickness were measured using a micro gauge (Mitutoyo, Kawasaki, Japan). The dissolution of the tablet was studied in 900 mL of phosphate buffer pH 7.4 with a paddle apparatus at 37 ± 0.5 °C and 100 rpm for 60 min (Agilent Technologies, Santa Clara, CA, USA). At predetermined time intervals, a 5 mL sample was withdrawn, filtered through a 0.45 μm PTFE-H filter, appropriately diluted, and analyzed using an HPLC system (LC-20AD, Shimadzu, Kyoto, Japan). The tablet samples were stored in a glass vial, sealed, and kept in a stability chamber (Jeio tech, Incheon, Korea) maintained at 40 °C and 75% RH for 30-day, and the assay and dissolution test were carried out. A sink condition was maintained throughout the dissolution study. The release profile was calculated with respect to the GLN Form I (reference). All data represent the means and standard deviation of six samples.

### 3.12. Design of Experiment and Statistical Analysis

The design of experiment was performed using Design Expert 11 software (Design Expert Inc., Minneapolis, MN, USA). The data are expressed as mean ± standard deviation (SD). The statistical analysis was performed using Origin 2018 software (OriginLab Co., Ltd., Northampton, MA, USA). Comparisons of the means were carried out using a paired *t*-test. A *p*-value < 0.05 (*) was considered to be statistically significant. 

## 4. Conclusions

The GLN solubility in DMSO, NMP, 1,4-dioxane, PEG 400, THP, and water was discussed. A higher solubility was observed in 1,4-dioxane, DMSO, NMP, and their respective aqueous mixtures. Modified AM (*RMSD* 0.124) and CNIBS/R-K model (*RMSD* 0.358) indicated a good agreement between the experimental and calculated data of GLN solubility in mono-solvent and binary solvent, respectively. KAT-LSER indicated GLN solubility could be influenced by the interactions of the solvent with the solute (decreased hydrogen bonding acidity and increased hydrogen bond basicity), increased electrostatic solute–solvent interactions. The activity coefficient and thermodynamic studies suggested that 1,4-dioxane could be the solvent for a recrystallization process. The dioxanate, DMSOte, NMPate, and THPate solvate crystals of GLN were obtained from the recrystallization process. The in vitro release profile after 30-day suggested that GLN-dioxanate had better release profile (93.74%) and stability compared to other studied solvates. The solubility determination of GLN and its application in the recrystallization process was very useful in identifying the most stable solvate which does not transform during the high-shear wet granulation and freeze-drying processes and had better in vitro release profile.

## Figures and Tables

**Figure 1 molecules-27-01392-f001:**
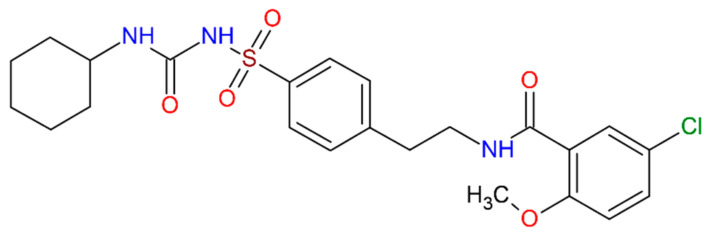
Molecular structure of the model drug, glibenclamide (GLN).

**Figure 2 molecules-27-01392-f002:**
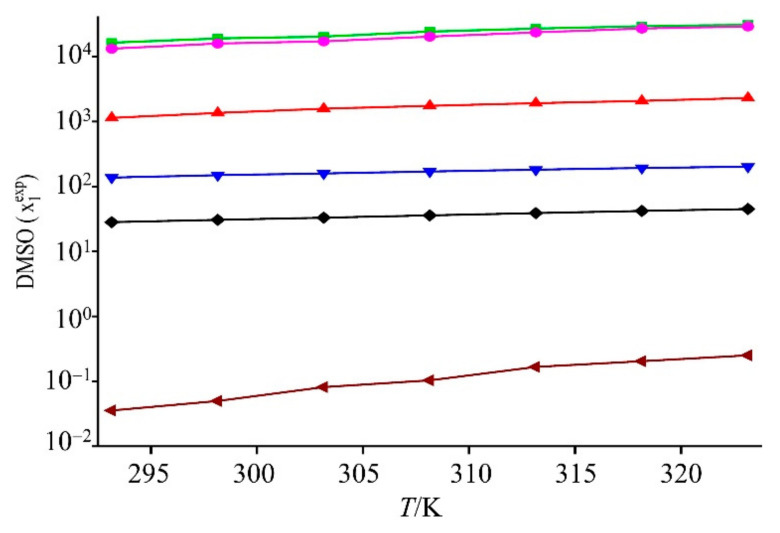
Experimental mole fraction solubility (*X*_exp_) of GLN in various organic solvents at *T* = (293.15~323.15) K: mole fraction solubility in DMSO (in green), NMP (in pink), 1,4-dioxane (in red), PEG 400 (in blue), THP (in black), and water (in maroon) are from static equilibrium solubility.

**Figure 3 molecules-27-01392-f003:**
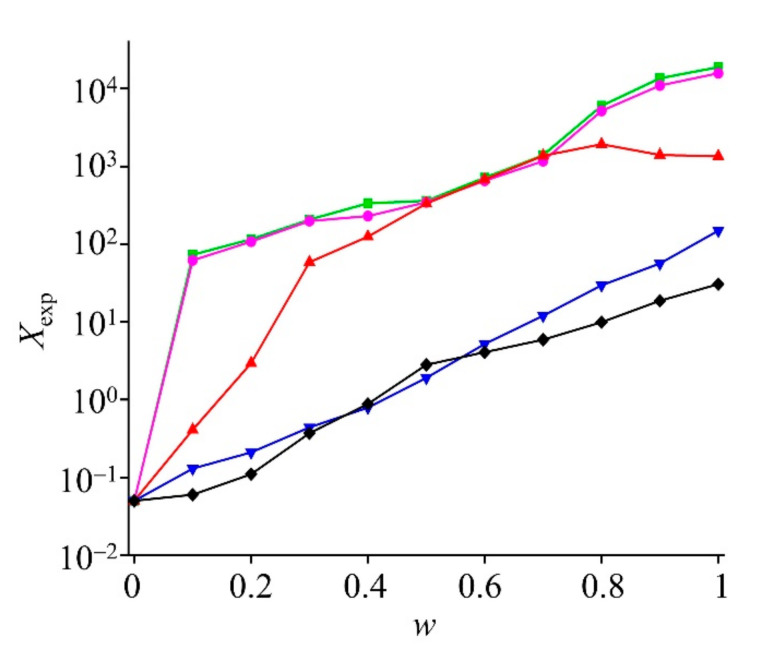
Experimental mole fraction solubility (*X*_exp_) of GLN in the mixtures of (DMSO + water, green line), (NMP + water, pink line), (1,4-dioxane + water, red line), (PEG 400 + water, blue line), and (THP + water, black line) at *T* = 298.15 K. Solvent is represented by *w* and water is represented by (1 – *w)*.

**Figure 4 molecules-27-01392-f004:**
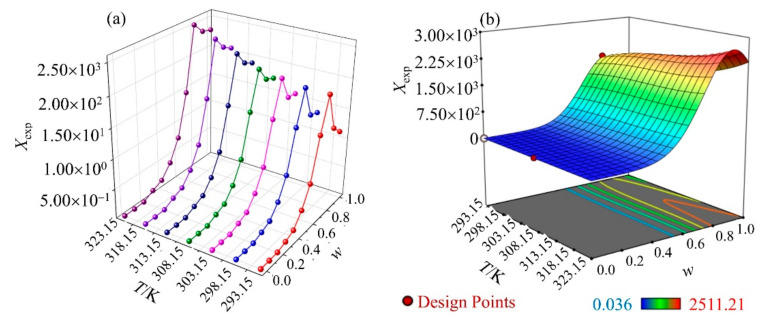
(**a**) Experimental mole fraction solubility (*X*_exp_) of GLN in a (1,4-dioxane + water) mixture at *T* = (293.15~323.15) K. (**b**) Optimization of mole fraction solubility of GLN vs *T*/K and mole fraction of 1,4-dioxane (*w*) in the (1,4-dioxane + water) mixture.

**Figure 5 molecules-27-01392-f005:**
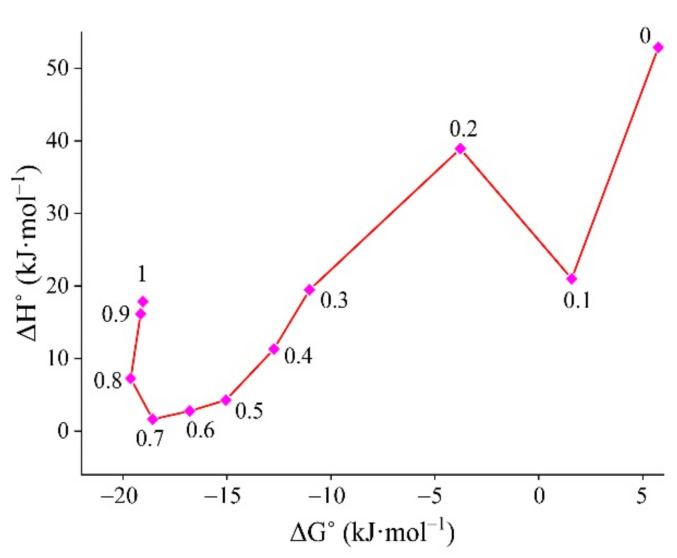
Enthalpy–entropy compensation analysis in different mole fractions of 1,4-dioxane (*w*) in (1,4-dioxane + water) mixture at a *T*_hm_ = 308.15 K. Mole fraction of 1,4-dioxane (*w*) in a (1,4-dioxane + water) mixture from 0 to 1.

**Figure 6 molecules-27-01392-f006:**
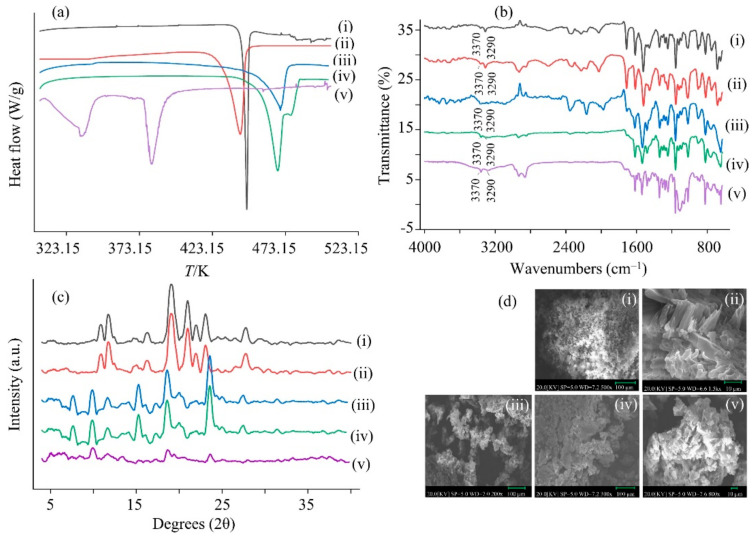
(**a**) DSC thermograms, (**b**) FTIR spectra, and (**c**) PXRD patterns of (i) GLN reference (in black), (ii) GLN-dioxanate (in red), (iii) GLN-NMPate (in blue), (iv) GLN-DMSOte (in green), and (v) GLN-THPate (in violet), respectively, and (**d**) SEM images of (i) GLN reference (×500), (ii) GLN-dioxanate (×1500), (iii) GLN-NMPate (×200), (iv) GLN-DMSOte (×300), and (v) GLN-THPate (×800).

**Figure 7 molecules-27-01392-f007:**
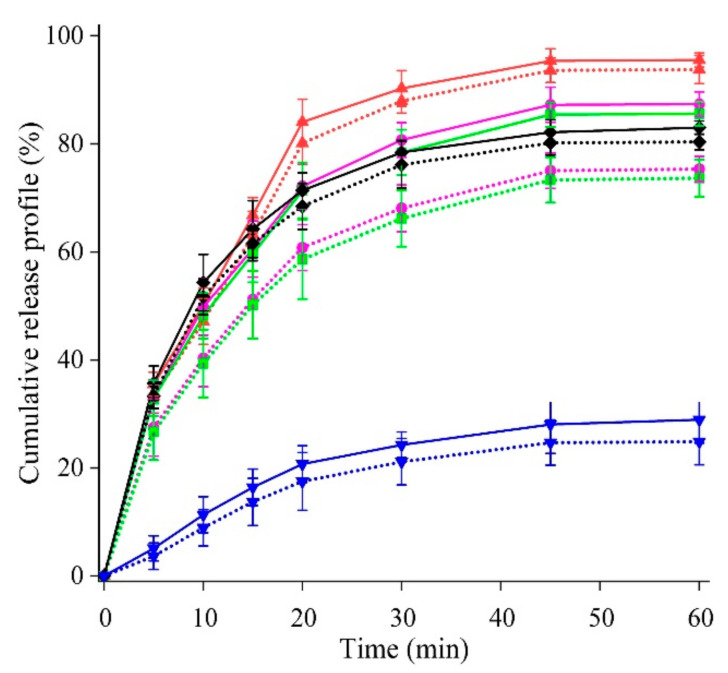
Cumulative release profile of the compressed tablets, formed from GLN reference at 0-day (in solid blue) and at 30-day (in dotted blue), GLN-dioxanate at 0 day (in solid red) and at 30-day (in dotted red), GLN-NMPate at 0 day (in solid pink) and at 30-day (dotted pink), GLN-DMSOte at 0 day (in solid green) and at 30-day (in dotted green), GLN-THPate at 0 day (in solid black) and at 30-day (in dotted black).

**Table 1 molecules-27-01392-t001:** Experimental and calculated mole fraction solubility of the GLN in organic solvents at *T* = (293.15~323.15) K.

*T*/K	$X_{exp}$ (Experimental)	*X*_AM_ (Modified AM)	*X*_Idl_ (Ideal Model)	*X*_λh_ (*λh* Model)
DMSO				
293.15	1.63 × 10^4^	1.62 × 10^4^	1.63 × 10^4^	1.30 × 10^4^
298.15	1.89 × 10^4^	1.87 × 10^4^	1.89 × 10^4^	1.54 × 10^4^
303.15	2.02 × 10^4^	2.12 × 10^4^	2.02 × 10^4^	1.81 × 10^4^
308.15	2.41 × 10^4^	2.38 × 10^4^	2.41 × 10^4^	2.13 × 10^4^
313.15	2.69 × 10^4^	2.63 × 10^4^	2.69 × 10^4^	2.49 × 10^4^
318.15	2.91 × 10^4^	2.88 × 10^4^	2.91 × 10^4^	2.90 × 10^4^
323.15	3.08 × 10^4^	3.12 × 10^4^	3.08 × 10^4^	3.38 × 10^4^
NMP				
293.15	1.32 × 10^4^	1.33 × 10^4^	1.32 × 10^4^	1.37 × 10^4^
298.15	1.58 × 10^4^	1.54 × 10^4^	1.58 × 10^4^	1.43 × 10^4^
303.15	1.71 × 10^4^	1.78 × 10^4^	1.71 × 10^4^	1.15 × 10^4^
308.15	2.02 × 10^4^	2.04 × 10^4^	2.02 × 10^4^	2.18 × 10^4^
313.15	2.35 × 10^4^	2.32 × 10^4^	2.35 × 10^4^	2.66 × 10^4^
318.15	2.70 × 10^4^	2.63 × 10^4^	2.70 × 10^4^	2.74 × 10^4^
323.15	2.90 × 10^4^	2.96 × 10^4^	2.90 × 10^4^	2.65 × 10^4^
1,4-Dioxane				
293.15	1.13 × 10^3^	1.15 × 10^3^	1.14 × 10^3^	1.18 × 10^3^
298.15	1.35 × 10^3^	1.35 × 10^3^	1.35 × 10^3^	1.15 × 10^3^
303.15	1.57 × 10^3^	1.55 × 10^3^	1.57 × 10^3^	1.68 × 10^3^
308.15	1.74 × 10^3^	1.74 × 10^3^	1.74 × 10^3^	1.75 × 10^3^
313.15	1.91 × 10^3^	1.93 × 10^3^	1.91 × 10^3^	1.99 × 10^3^
318.15	2.08 × 10^3^	2.11 × 10^3^	2.07 × 10^3^	2.15 × 10^3^
323.15	2.30 × 10^3^	2.27 × 10^3^	2.30 × 10^3^	2.39 × 10^3^
PEG 400				
293.15	1.37 × 10^2^	1.35 × 10^2^	1.36 × 10^2^	1.34 × 10^2^
298.15	1.49 × 10^2^	1.46 × 10^2^	1.48 × 10^2^	1.47 × 10^2^
303.15	1.58 × 10^2^	1.55 × 10^2^	1.57 × 10^2^	1.55 × 10^2^
308.15	1.70 × 10^2^	1.66 × 10^2^	1.69 × 10^2^	1.67 × 10^2^
313.15	1.81 × 10^2^	1.78 × 10^2^	1.80 × 10^2^	1.78 × 10^2^
318.15	1.93 × 10^2^	1.94 × 10^2^	1.92 × 10^2^	1.89 × 10^2^
323.15	2.01 × 10^2^	2.04 × 10^2^	2.00 × 10^2^	1.97 × 10^2^
THP				
293.15	2.82 × 10^1^	2.81 × 10^1^	2.72 × 10^1^	2.72 × 10^1^
298.15	3.07 × 10^1^	3.01 × 10^1^	2.97 × 10^1^	2.96 × 10^1^
303.15	3.31 × 10^1^	3.28 × 10^1^	3.21 × 10^1^	3.21 × 10^1^
308.15	3.59 × 10^1^	3.51 × 10^1^	3.49 × 10^1^	3.49 × 10^1^
313.15	3.89 × 10^1^	3.90 × 10^1^	3.79 × 10^1^	3.78 × 10^1^
318.15	4.20 × 10^1^	4.22 × 10^1^	4.10 × 10^1^	4.09 × 10^1^
323.15	4.49 × 10^1^	4.53 × 10^1^	4.40 × 10^1^	4.40 × 10^1^
Water				
293.15	3.57 × 10^−2^	3.45 × 10^−2^	3.45 × 10^−2^	3.24 × 10^−2^
298.15	5.00 × 10^−2^	5.32 × 10^−2^	5.32 × 10^−2^	4.91 × 10^−2^
303.15	8.16 × 10^−2^	7.87 × 10^−2^	7.87 × 10^−2^	7.96 × 10^−2^
308.15	1.04 × 10^−1^	1.12 × 10^−1^	1.12 × 10^−1^	1.01 × 10^−1^
313.15	1.67 × 10^−1^	1.53 × 10^−1^	1.53 × 10^−1^	1.61 × 10^−1^
318.15	2.04 × 10^−1^	2.02 × 10^−1^	2.02 × 10^−1^	2.01 × 10^−1^
323.15	2.51 × 10^−1^	2.52 × 10^−1^	2.58 × 10^−1^	2.59 × 10^−1^

Standard uncertainties, *u(T)* = 0.05, *u(X)* = 1.08.

**Table 2 molecules-27-01392-t002:** Application of Fedors’ method for estimating the internal energy, molar volume, and Hildebrand solubility parameter of GLN.

Group	Group Number	$\Delta_{ei}\;(\mathrm{kJ}\cdot \mathrm{mol}{}^{-1})$	$\Delta_{vi}\;({\mathrm{cm}}^{3}\cdot \mathrm{mol}{}^{-1})$
CONH	1	33.47	9.5
CH_2_	2	2 × 4.94 = 9.88	2 × 16.1 = 32.2
Cl	1	11.55	24
S	1	14.14	12
NH	2	2 × 8.37 = 16.74	2 × 4.5 = 9
O	4	4 × 3.35 = 13.40	4 × 3.8 = 15.2
Phenylene (p)	1	31.92	52.4
Phenyl (tri-substituted)	1	31.92	33.4
6-member ring closure	1	1.05	16
Total		164.07	203.7
Solubility parameter	(164,070/203.7)^1/2^ = 28.38 MPa^1/2^		

**Table 3 molecules-27-01392-t003:** Solvatochromic parameters (*α*, *β*, and *π**) and Hildebrand solubility parameter (*δ*_H_) for solvents (referred to Table 1 in Jessop et al. [28]).

Solvent	*α*	*β*	*π**	*δ*_H_ (MPa^1/2^)
DMSO	0.00	−	0.94	13.00
NMP	0.00	0.77	0.92	23.10
1,4-Dioxane	0.00	0.37	0.55	23.40
PEG 400	0.31	0.75	0.91	32.90
THP	0.00	−	0.64	22.30
Water	1.17	0.47	1.09	47.82

**Table 4 molecules-27-01392-t004:** Parameters of the modified Apelblat equation, Ideal equation, and *λh* equation for GLN in organic solvents.

	Modified Apelblat Model					Ideal Model				*λh* Model			
Solvent	*A**	10^3^ *B**	*C*	10^4^ *RMSD*	10^2^ *MRD*	*A^#^*	10^3^ *B**^#^*	10^4^ *RMSD*	10^2^ *MRD*	*λ*	10^3^*h*	10^4^ *RMSD*	10^2^ *MRD*
DMSO	236.67	−12.12	−32.68	0.189	0.001	1.03	33.46	0.392	0.239	−0.32	−8448.28	2.426	1.915
NMP	70.70	−4.94	−7.81	0.032	0.281	9.13	296.78	0.379	0.738	−0.56	−1313	2.155	1.179
Dioxane	351.50	−17.56	−50.09	0.078	0.164	7.49	2.43	0.231	0.612	−0.16	−968.61	0.178	1.824
PEG 400	54.97	−2.67	−7.73	0.086	13.654	722.19	2.35	0.227	0.857	8.32	−8969.38	1.703	0.993
THP	54.97	−2.67	−7.73	0.132	11.300	151.21	0.49	1.072	4.052	35.00	−3660.41	3.566	0.971
Water	709.79	−37.97	−102.74	0.224	0.684	372.41	0.01	2.079	6.837	72.44	100.30	0.847	0.598
Overall	0.124				4.347			0.730	2.223			1.813	1.247

Relative uncertainties, *u(A*)* = 3.51, *u(B*)* = 1.74, *u(C)* = 1.06, *u(A^#^)* = 2.11, *u(B^#^)* = 3.95, *u(λ)* = 1.28, *u(h)* = 1.03.

**Table 5 molecules-27-01392-t005:** Experimental (*X*_exp_) and calculated (*X*_AM_, *X*_IDL_, *X*_RK_, and *X*_JA_) mole fraction solubility of the GLN in 1,4-dioxane (*w*) + water (1 − *w*) mixture at *T* = (293.15~323.15) K.

*T*/K	$X_{exp}$	*X*_AM_ (Modified AM)	*X*_IDL_ (Ideal Model)	*X*_RK_ (CNIBS/R-K)	*X*_JA_ (Modified JA)
*w* = 0.1					
293.15	3.79 × 10^−1^	3.75 × 10^−1^	3.74 × 10^−1^	3.75 × 10^−1^	3.96 × 10^−1^
298.15	4.10 × 10^−1^	4.32 × 10^−1^	4.32 × 10^−1^	4.32 × 10^−1^	4.86 × 10^−1^
303.15	4.71 × 10^−1^	4.25 × 10^−1^	4.78 × 10^−1^	4.75 × 10^−1^	5.27 × 10^−1^
308.15	5.12 × 10^−1^	5.18 × 10^−1^	5.11 × 10^−1^	5.18 × 10^−1^	5.65 × 10^−1^
313.15	5.91 × 10^−1^	5.15 × 10^−1^	6.02 × 10^−1^	5.95 × 10^−1^	6.01 × 10^−1^
318.15	7.11 × 10^−1^	7.13 × 10^−1^	7.15 × 10^−1^	7.13 × 10^−1^	7.28 × 10^−1^
323.15	8.47 × 10^−1^	8.11 × 10^−1^	8.58 × 10^−1^	8.41 × 10^−1^	8.59 × 10^−1^
*w* = 0.2					
293.15	2.51	2.08	2.53	2.67	2.36
298.15	2.95	2.78	2.94	2.92	2.99
303.15	3.38	3.21	3.49	3.30	3.65
308.15	4.03	3.39	4.12	4.02	4.09
313.15	4.57	4.51	4.71	4.68	4.80
318.15	5.13	5.62	5.61	5.18	5.17
323.15	5.71	5.74	5.70	5.69	5.59
*w* = 0.3					
293.15	4.98 × 10^1^	4.90 × 10^1^	4.72 × 10^1^	4.72 × 10^1^	4.49 × 10^1^
298.15	5.86 × 10^1^	5.37 × 10^1^	5.88 × 10^1^	5.97 × 10^1^	5.71 × 10^1^
303.15	6.72 × 10^1^	6.20 × 10^1^	7.01 × 10^1^	6.51 × 10^1^	6.65 × 10^1^
308.15	8.00 × 10^1^	8.38 × 10^1^	8.09 × 10^1^	8.06 × 10^1^	8.22 × 10^1^
313.15	9.08 × 10^1^	9.51 × 10^1^	9.14 × 10^1^	9.20 × 10^1^	9.61 × 10^1^
318.15	1.02 × 10^2^	1.06 × 10^2^	1.01 × 10^2^	1.01 × 10^2^	1.17 × 10^2^
323.15	1.13 × 10^2^	1.30 × 10^2^	1.11 × 10^2^	1.14 × 10^2^	1.35 × 10^2^
*w* = 0.4					
293.15	1.07 × 10^2^	1.26 × 10^2^	1.01 × 10^2^	1.06 × 10^2^	9.81 × 10^1^
298.15	1.25 × 10^2^	1.38 × 10^2^	1.26 × 10^2^	1.27 × 10^2^	1.14 × 10^2^
303.15	1.44 × 10^2^	1.67 × 10^2^	1.49 × 10^2^	1.56 × 10^2^	1.38 × 10^2^
308.15	1.71 × 10^2^	1.83 × 10^2^	1.73 × 10^2^	1.68 × 10^2^	1.89 × 10^2^
313.15	1.94 × 10^2^	1.96 × 10^2^	1.95 × 10^2^	1.96 × 10^2^	1.97 × 10^2^
318.15	2.18 × 10^2^	2.14 × 10^2^	2.17 × 10^2^	2.55 × 10^2^	1.99 × 10^2^
323.15	2.42 × 10^2^	2.49 × 10^2^	2.38 × 10^2^	2.70 × 10^2^	2.17 × 10^2^
*w* = 0.5					
293.15	2.85 × 10^2^	2.81 × 10^2^	2.69 × 10^2^	2.81 × 10^2^	2.48 × 10^2^
298.15	3.34 × 10^2^	3.65 × 10^2^	3.35 × 10^2^	3.42 × 10^2^	3.43 × 10^2^
303.15	3.83 × 10^2^	3.94 × 10^2^	3.99 × 10^2^	3.46 × 10^2^	3.94 × 10^2^
308.15	4.56 × 10^2^	4.47 × 10^2^	4.61 × 10^2^	4.75 × 10^2^	4.37 × 10^2^
313.15	5.18 × 10^2^	5.65 × 10^2^	5.21 × 10^2^	5.19 × 10^2^	5.16 × 10^2^
318.15	5.81 × 10^2^	5.94 × 10^2^	5.79 × 10^2^	5.80 × 10^2^	5.97 × 10^2^
323.15	6.46 × 10^2^	6.36 × 10^2^	6.35 × 10^2^	6.27 × 10^2^	6.48 × 10^2^
*w* = 0.6					
293.15	5.78 × 10^2^	6.35 × 10^2^	5.47 × 10^2^	5.72 × 10^2^	5.86 × 10^2^
298.15	6.79 × 10^2^	6.83 × 10^2^	6.81 × 10^2^	6.49 × 10^2^	6.72 × 10^2^
303.15	7.79 × 10^2^	7.95 × 10^2^	8.11 × 10^2^	7.61 × 10^2^	7.47 × 10^2^
308.15	9.27 × 10^2^	9.07 × 10^2^	9.37 × 10^2^	9.54 × 10^2^	9.03 × 10^2^
313.15	1.05 × 10^3^	1.21 × 10^3^	1.06 × 10^3^	1.02 × 10^3^	1.27 × 10^3^
318.15	1.18 × 10^3^	1.24 × 10^3^	1.17 × 10^3^	1.17 × 10^3^	1.52 × 10^3^
323.15	1.31 × 10^3^	1.35 × 10^3^	1.29 × 10^3^	1.29 × 10^3^	1.27 × 10^3^
*w* = 0.7					
293.15	1.17 × 10^3^	1.06 × 10^3^	1.11 × 10^3^	1.16 × 10^3^	1.14 × 10^3^
298.15	1.37 × 10^3^	1.22 × 10^3^	1.38 × 10^3^	1.34 × 10^3^	1.23 × 10^3^
303.15	1.58 × 10^3^	1.36 × 10^3^	1.64 × 10^3^	1.57 × 10^3^	1.46 × 10^3^
308.15	1.88 × 10^3^	1.53 × 10^3^	1.90 × 10^3^	1.82 × 10^3^	1.95 × 10^3^
313.15	2.13 × 10^3^	1.97 × 10^3^	2.15 × 10^3^	2.19 × 10^3^	2.18 × 10^3^
318.15	2.39 × 10^3^	2.34 × 10^3^	2.38 × 10^3^	2.38 × 10^3^	2.31 × 10^3^
323.15	2.66 × 10^3^	2.58 × 10^3^	2.62 × 10^3^	2.65 × 10^3^	2.82 × 10^3^
*w* = 0.8					
293.15	1.64 × 10^3^	1.41 × 10^3^	1.56 × 10^3^	1.65 × 10^3^	1.75 × 10^3^
298.15	1.93 × 10^3^	1.87 × 10^3^	1.94 × 10^3^	2.01 × 10^3^	2.01 × 10^3^
303.15	2.21 × 10^3^	1.99 × 10^3^	2.31 × 10^3^	2.26 × 10^3^	2.27 × 10^3^
308.15	2.63 × 10^3^	2.56 × 10^3^	2.67 × 10^3^	2.73 × 10^3^	2.82 × 10^3^
313.15	2.99 × 10^3^	3.00 × 10^3^	3.01 × 10^3^	3.04 × 10^3^	3.10 × 10^3^
318.15	3.35 × 10^3^	3.11 × 10^3^	3.35 × 10^3^	3.37 × 10^3^	3.21 × 10^3^
323.15	3.74 × 10^3^	3.73 × 10^3^	3.67 × 10^3^	3.73 × 10^3^	3.53 × 10^3^
*w* = 0.9					
293.15	1.19 × 10^3^	1.08 × 10^3^	1.13 × 10^3^	1.20 × 10^3^	1.08 × 10^3^
298.15	1.40 × 10^3^	1.38 × 10^3^	1.41 × 10^3^	1.41 × 10^3^	1.29 × 10^3^
303.15	1.61 × 10^3^	1.68 × 10^3^	1.67 × 10^3^	1.63 × 10^3^	1.59 × 10^3^
308.15	1.92 × 10^3^	1.91 × 10^3^	1.94 × 10^3^	1.80 × 10^3^	1.95 × 10^3^
313.15	2.17 × 10^3^	2.24 × 10^3^	2.19 × 10^3^	2.15 × 10^3^	2.14 × 10^3^
318.15	2.43 × 10^3^	2.38 × 10^3^	2.44 × 10^3^	2.49 × 10^3^	2.51 × 10^3^
323.15	2.71 × 10^3^	2.79 × 10^3^	2.67 × 10^3^	2.75 × 10^3^	2.66 × 10^3^

Standard uncertainties, *u(T)* = 0.05, *u(X)* = 0.66.

**Table 6 molecules-27-01392-t006:** Parameters of the modified Apelblat equation and Ideal equation for GLN in a 1,4-dioxane (*w*) + water (1 − *w*) mixture.

	Modified Apelblat Model					Ideal Model			
*w*	*A**	10^2^ *B****	*C*	10^4^ *RMSD*	10^2^ *MRD*	10^3^ *A**^#^*	*B^#^*	10^4^ *RMSD*	10^2^ *MRD*
0	709.79	−3.80	−102.74	0.781	0.153	−0.70	2.40	0.015	0.165
0.1	1.28	9.41	−0.58	0.144	0.471	1.03	-3.09	0.096	0.443
0.2	−5.44	−3.32	0.97	1.126	0.945	−10.22	37.24	0.085	0.794
0.3	−2.45	−1.18	0.96	0.291	1.356	−202.91	739.40	0.168	1.197
0.4	−1.69	−4.41	1.02	0.629	1.382	−433.96	1581.32	0.359	2.833
0.5	−0.71	1.67	0.99	1.274	0.838	−1157.55	4218.03	0.958	1.461
0.6	1.02	2.64	1.04	0.347	0.412	−2352.07	8570.75	1.419	1.947
0.7	0.71	−3.55	0.97	0.409	0.378	−4761.96	17,352.21	0.935	3.942
0.8	1.05	6.48	1.02	0.406	0.752	−6690.15	24,378.38	0.872	5.539
0.9	0.73	−8.14	0.98	0.221	0.428	−4863.29	17,721.47	0.634	4.432
1	5.71	4.23	0.33	0.102	0.213	−3570.12	13,325.61	1.888	1.655
Overall				0.519	0.664			0.675	2.218

Relative uncertainties, u(*A**) = 2.19, u(*B**) = 1.41, u(*C*) = 1.07, u(*A^#^*) = 2.49, u(*B^#^*) = 1.32.

**Table 7 molecules-27-01392-t007:** Parameters of the CNIBS/R-K model and Jouyban–Acree model for GLN in a 1,4-dioxane (*w*) + water (1 − *w*) mixture at *T* = (293.15~323.15) K.

CNIBS/R-K Model								
*T*/K	*B* _0_	*B* _1_	*B* _2_	*B* _3_	*B* _4_	10^4^*RMSD*	10^2^*MRD*	
293.15	−0.9071	1.4012	−1.2666	3.2724	−2.0394	0.229	2.523	
298.15	−0.9906	1.6046	−1.4614	3.7922	−2.3645	0.271	2.986	
303.15	−1.0535	1.7962	−1.6484	4.2956	−2.6798	0.314	3.451	
308.15	−1.8161	2.4434	−2.1569	5.4956	−3.4195	0.359	3.946	
313.15	−2.3361	2.9271	−2.5459	6.4286	−3.9957	0.400	4.396	
318.15	−2.8899	3.4332	−2.9511	7.3975	−4.5939	0.441	4.854	
323.15	−3.2435	3.8378	−3.2956	8.2557	−5.1265	0.491	5.396	
Overall						0.358	3.936	
Modified Jouyban–Acree Model								
*A* _1_	*A* _2_	*A* _3_	*A* _4_	*A* _5_	*A* _6_	10^4^*A*_7_	*A* _8_	*A* _9_
−7.2675	−260.9930	−0.4723	82.1708	255.7966	−677.5864	1.7237	−1072.8059	143.9475
10^4^*RMSD* = 0.6891								
10^2^*MRD =* 4.2473								

Standard uncertainty, *u*(*T*) = 0.05.

**Table 8 molecules-27-01392-t008:** Activity coefficients (*γ*_i_) of GLN in various solvents at *T* = (293.15~323.15) K.

Solvents	*γ* _i_						
	293.15 K	298.15 K	303.15 K	308.15 K	313.15 K	318.15 K	323.15 K
DMSO	2.80	1.36	7.40 × 10^−1^	3.66 × 10^−1^	1.98 × 10^−1^	1.13 × 10^−1^	6.66 × 10^−2^
NMP	3.46	1.63	8.73 × 10^−1^	4.37 × 10^−1^	2.27 × 10^−1^	1.21 × 10^−1^	7.08 × 10^−2^
Dioxane	4.01 × 10^1^	1.91 × 10^1^	9.49	5.07	2.79	1.58	8.91 × 10^−1^
PEG 400	3.33 × 10^2^	1.73 × 10^2^	9.43 × 10^1^	5.19 × 10^1^	2.94 × 10^1^	1.70 × 10^1^	1.02 × 10^1^
THP	1.62 × 10^3^	8.41 × 10^2^	4.51 × 10^2^	2.46 × 10^2^	1.37 × 10^2^	7.81 × 10^1^	4.56 × 10^1^
Water	1.28 × 10^6^	5.16 × 10^5^	1.83 × 10^5^	8.52 × 10^4^	3.19 × 10^4^	1.61 × 10^4^	8.18 × 10^3^

Standard uncertainty, *u*(*T*) = 0.05.

**Table 9 molecules-27-01392-t009:** Apparent thermodynamic parameters for the dissolution behavior of GLN in different solvents.

Solvent	Δ*H°*	Δ*G*°	Δ*S*°	*R* ^2^
	kJ·mol^−1^	kJ·mol^−1^	J·mol^−1^·K^−1^	
DMSO	−10.04	−2.30	−25.12	0.9992
NMP	−11.56	−1.96	−31.17	0.9996
Dioxane	−9.65	4.42	−45.70	0.9992
PEG 400	−5.52	10.38	−51.60	0.9990
THP	−6.17	14.33	−66.57	0.9993
Water	30.33	28.96	4.44	0.9996

Standard uncertainty, *u*(*T*) = 0.05.

**Table 10 molecules-27-01392-t010:** Apparent thermodynamic parameters for the dissolution behavior of GLN in a 1,4-dioxane (*w*) + water (1 − *w*) mixture.

*w*	Δ*H*°	Δ*G*°	Δ*S*°	*R* ^2^
	kJ·mol^−1^	kJ·mol^−1^	J·mol^−1^ K^−1^	
0	52.89	5.75	153.08	0.9996
0.1	20.99	1.58	63.03	0.9991
0.2	38.93	−3.77	138.64	0.9990
0.3	19.47	−11.03	99.02	0.9995
0.4	11.33	−12.74	78.15	0.9992
0.5	4.28	−15.05	62.76	0.9991
0.6	2.77	−16.79	63.49	0.9996
0.7	1.65	−18.57	65.63	0.9998
0.8	7.26	−19.63	87.32	0.9990
0.9	16.15	−19.15	114.63	0.9990
1	17.85	−19.04	119.79	0.9992

Standard uncertainties, *u*(*T*) = 0.05.

## Data Availability

The data underlying in this article will be shared on reasonable request to the corresponding author.

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
