# Peer review of "Correlation of Solubility Thermodynamics of Glibenclamide with Recrystallization and In Vitro Release Profile"

_molecules, 2022, doi:10.3390/molecules27041392_

Round 1

Reviewer 1 Report

The manuscript molecules-1595626 presents a study of solubility of glibenclamide in various solvents. The preparation of a soluble form of glibenclamide is of both scientific and commercial interest to the pharmaceutical industry. However, this manuscript contains mostly only a statement of the results, some of which are simply consistent with previous studies, but very little discussion of the results, and in section 2.2 and for most of the tables there is no discussion at all. The agreement with previous studies is certainly good, but then what is original about this manuscript? Although in the Conclusions section, the authors write that "The GLN solubility in different solvents was discussed." In addition to the lack of a clear discussion of the results obtained, there are the following questions and comments:

  1. It is not clear what is the reason for the choice of toxic organic solvents to increase the solubility of glibenclamide? Of the presented solvents, only PEG 400 can be suitable for creating oral forms.
  2. In the Introduction section, the sentence about various strategies to improve the dissolution of glibenclamide should be supplemented with the use of cyclodextrins.
  3. What does "most consistent in vitro release profile" mean?
  4. Is it appropriate to use the term “release” if glibenclamide was not encapsulated anywhere in the presented work?
  5. In the conclusion, the authors state that according to one model, "the increment in GLN solubility could be due to the interactions of hydrogen bond basicity/acidity and electrostatic solvent-solvent interactions." The authors used a whole range of research methods, and this conclusion is based on only one model. Why did the results of, for example, IR spectroscopy fail to confirm the presence of hydrogen bonds and electrostatic interactions?

Author Response

Reviewer #1: The manuscript molecules-1595626 presents a study of solubility of glibenclamide in various solvents. The preparation of a soluble form of glibenclamide is of both scientific and commercial interest to the pharmaceutical industry. However, this manuscript contains mostly only a statement of the results, some of which are simply consistent with previous studies, but very little discussion of the results, and in section 2.2 and for most of the tables there is no discussion at all.

The authors really appreciate the comments and recommendations that the reviewer provided for the manuscript. The reviewer’s input has been invaluable to the authors during the revision process. Responses to each comment are updated in the manuscript and highlighted in yellow.

The authors hoped that the present study would influence the readers and motivate them to pay more attention on solid state form. As asked, additional efforts are made to improve the quality of the manuscript: discussion section 2.2 (line 245, line 251) and the tables (line 114 – Table 1; line 153 – Table 2; line 154 – Table 3; line 175 – Table 4; line 197 – Table 5; line 228 – Table 6, Table 7; line 251 – Table 8; line 274 and 285 – Table 9;  line 297 – Table 10) are updated in the manuscript.

The agreement with previous studies is certainly good, but then what is original about this manuscript? Although in the Conclusions section, the authors write that "The GLN solubility in different solvents was discussed."

The authors really appreciate the reviewer’s comment. The effort is to find the balance between the increased dissolution rate (line 398) with respect to various solubility data based on different crystalline solvates. Moreover, higher degree of glibenclamide purity could be obtained (line 380). Connecting the solvate/polymorph forms with decent material properties to drug product development would be necessary for better exposure and patient compliance as well. The authors hope the reviewer consider the points.

As the reviewer suggested, the authors updated the discussion section to support the conclusion statement i.e., smaller RMSD value (0.124) of modified AM model (line 175), smaller RMSD value (0.358) of CNIBS/R-K model (line 228), decreased activity coefficients (gi) with respect to rise in temperature (line 245), GLN solubility in water depending on temperature (  = 30.33, line 274), and lower values of thermodynamic parameters ( , ,  , line 297) in relation to improved solubility. The conclusion section is updated accordingly (line 627).

In addition to the lack of a clear discussion of the results obtained, there are the following questions and comments:

  1. It is not clear what is the reason for the choice of toxic organic solvents to increase the solubility of glibenclamide? Of the presented solvents, only PEG 400 can be suitable for creating oral forms.

The authors agree with the reviewer’s comments that the selected solvents were of class 2 (line 74) and class 3 (line 75). The authors believe that during the synthesis of drug compound, an isocyanate is used in the existing synthetic route, which is a toxic compound. Recently published article [https://doi.org/10.1002/chem.202103196] suggested a safer molecule N-carbamate in its place during the synthetic route. Although, glibenclamide produced may be safer but still had low dissolution in its polymorphic form (Form I). Therefore, the efforts were to study the solvate forms, which could add one more step on the existing polymorphic form (Form I), as discussed in section 3.6 (line 536). The desired characteristics were higher degree of glibenclamide purity in the tablets with respect to impurities and degradation products (line 380) and the improved release profile (line 398), in which the present study had achieved. After the successful discovery of the solvate form, the purity of the drug compound can be worked upon to the limits allowed by the ICH Q3C(R6) guidelines. In stringent dosage form such as long-acting injectable microsphere, the organic solvents (chloroform, DCM, methanol) were used, which are reduced below 600 ppm (allowable limit, https://doi.org/10.1016/j.jddst.2021.102608) in the final product. The authors hope the current study may provide an idea when to improve solubility or release profiles without extensive formulation campaign.

  1. In the Introduction section, the sentence about various strategies to improve the dissolution of glibenclamide should be supplemented with the use of cyclodextrins.

As recommended, the introduction section (line 58) and additional reference were updated and highlighted in yellow.

  1. What does "most consistent in vitro release profile" mean?

The authors are thankful for the reviewer’s comment. The line is rephrased to ‘…gave higher cumulative release profile….’ (line 400) for the better understanding and the change is highlighted in yellow.

  1. Is it appropriate to use the term “release” if glibenclamide was not encapsulated anywhere in the presented work?

The authors appreciate the reviewer’s keen point. The glibenclamide was granulated using a high shear (line 596) and compressed into tablets (line 605) before in vitro dissolution study was observed. The authors would like to mention that many scientists are using ‘release’ or ‘dissolution’ without any detailed information as the reviewer pointed out. Since the authors evaluated tablets, ‘release’ might be used to tell how much drug is release out depending on time. If the reviewer consider that it is not still appropriate, please let the authors know and the authors will revise them.

  1. In the conclusion, the authors state that according to one model, "the increment in GLN solubility could be due to the interactions of hydrogen bond basicity/acidity and electrostatic solvent-solvent interactions." The authors used a whole range of research methods, and this conclusion is based on only one model.

The authors are thankful for the reviewer’s comment. The authors included all models in the conclusion section which were discussed (modified AM model, CNIBS/R-K model, line 629) along with KAT-LSER model.

Why did the results of, for example, IR spectroscopy fail to confirm the presence of hydrogen bonds and electrostatic interactions?

The authors agree with the reviewer’s comment that the hydrogen bonding and interactions of glibenclamide Form I with the solvates were not confirmed by the IR spectroscopy. The probable reason could be the incorporation of liquid solvent in the solid compound. The entrapped liquid solvents in GLN were not detected by IR spectroscopy. GLN in its solvate forms remained unchanged (Form I, characteristics absorption band at 3290 and 3370 cm-1, line 349). The increased release profile (higher solubility – Table 1), lower melting point (DSC thermogram – melting point of 443.16 K – line 321), and KAT-LSER model (Eq. 2 and line 165) suggested the characteristics of hydrogen bonding (line 321). The manuscript is updated and the changes are highlighted in yellow.

Reviewer 2 Report

The paper seems to be  well written but the description of the solvents used is not clear in the abstract. Also, the stability study is not clear and is not carried according to the quid lines for active pharmaceutical ingredients. Therefore, revision  is suggested in order to make clear the solvents used in the abstract and the stability study in the experimental  part.  

Author Response

Reviewer #2: The paper seems to be well written but the description of the solvents used is not clear in the abstract.

The authors really appreciate the comments and recommendations that the reviewer provided for the manuscript. The reviewer’s input has been invaluable to the authors during the revision process. Responses to each comment are updated in the manuscript and highlighted in yellow. As suggested, the abstract is rephrased (line 12) to provide the solvents and highlighted in yellow.

Also, the stability study is not clear and is not carried according to the quid lines for active pharmaceutical ingredients.

The authors appreciate the reviewer’s comment. The authors had conducted the stability of the GLN tablets in order to find the balance between the increased dissolution rate (line 398) and the higher degree of GLN purity in the tablets with respect to impurities and degradation products (line 380). The release profile was calculated with respect to the GLN Form I. The stress testing of active pharmaceutical ingredients at interval of 10 oC incremental temperature and the shelf-life extrapolation from the accelerated stability studies were not in the scope of the study (line 86). The section 2.5 (line 380) of the manuscript is updated for the ease of the readers and the changes are highlighted in yellow.

Therefore, revision is suggested in order to make clear the solvents used in the abstract and the stability study in the experimental part.

The authors are very thankful for the reviewer’s comment. As suggested, the abstract (line 13) and the stability study (line 86, line 380, line 615) are updated and highlighted in yellow.

Reviewer 3 Report

In the manuscript, the authors investigated the solubility of glibenclamide in various solvents. Their results were used to find the ideal solvent for the recrystallization process, creating the most stable solvate suitable for the wet granulation process. The authors report the wealth of measurement results in the manuscript. The figures and tables are very informative, the manuscript is well edited. Some small comments, questions: The authors took the type and proportion of excipients used for wet granulation from the 6th reference. However, in this article (Bonfilio et al .; doi: 10.1002 / jps.22799) direct compression is performed and the excipients were selected according to this process. Regardless, in the present manuscript, the first step in forming the tablet form is wet granulation. Were any excipients used, such as magnesium stearate, during granulation? What type of povidone was used? In my opinion, it would be good to indicate in the materials section the exact type, brand name, and manufacturer of the used excipients. Why was it necessary to freeze-drying for 72 hours? Was the moisture content examined during lyophilization because wet granulation occurred where water was the granulating liquid? Some machine types and manufacturers are not listed such as freeze-dryer or stability chamber. How was the calculation performed during the drug release study? What did the authors take 100% for each tablet? Has the theoretical content been calculated, or has the active ingredient content of the formulas been determined according to the pharmacopeia? 

Author Response

Reviewer #3: In the manuscript, the authors investigated the solubility of glibenclamide in various solvents. Their results were used to find the ideal solvent for the recrystallization process, creating the most stable solvate suitable for the wet granulation process.

The authors report the wealth of measurement results in the manuscript. The figures and tables are very informative, the manuscript is well edited.

The authors really appreciate the comments and recommendations that the reviewer provided for the manuscript. The reviewer’s input has been invaluable to the authors during the revision process. Responses to each comment are updated in the manuscript and highlighted in yellow.

Some small comments, questions: The authors took the type and proportion of excipients used for wet granulation from the 6th reference. However, in this article (Bonfilio et al .; doi: 10.1002 / jps.22799) direct compression is performed and the excipients were selected according to this process. Regardless, in the present manuscript, the first step in forming the tablet form is wet granulation.

The authors really appreciate the reviewer’s point. As suggested by the reviewer, the excipients and the brief process are included in the method section 3.10 of the manuscript (line 596) and the changes are highlighted in yellow. The reference is updated accordingly.

Were any excipients used, such as magnesium stearate, during granulation? What type of povidone was used? In my opinion, it would be good to indicate in the materials section the exact type, brand name, and manufacturer of the used excipients.

The authors are very thankful for the reviewer’s suggestion. The excipient details are included in the material section 3.1 (line 420) and the amount of excipient used are explained in method section 3.10 (line 596) as well.

Why was it necessary to freeze-drying for 72 hours? Was the moisture content examined during lyophilization because wet granulation occurred where water was the granulating liquid?

The authors are thankful for the reviewer’s comment. The granules were dried until the constant weight was achieved (line 600). The authors agree with the reviewer’s comment that the freeze-drying process has taken a longer time (72 hours) while attempting to maintain the moisture content to less than 1.5% w/w. The authors wanted to exclude any thermal effects on the stability, if any, and hope the reviewer consider the point.

Some machine types and manufacturers are not listed such as freeze-dryer or stability chamber.

As suggested, more information on the machines were updated (line 600 and line 615) and the changes are highlighted in yellow.

How was the calculation performed during the drug release study? What did the authors take 100% for each tablet? Has the theoretical content been calculated, or has the active ingredient content of the formulas been determined according to the pharmacopeia?

The authors are thankful for the reviewer’s comment. For the dissolution study, the tablets were prepared as provided in the section 3.10 containing 1.0% API, which was considered as 100% for each tablet. The dissolution profiles were compared to see any changes among the solvates.

Round 2

Reviewer 1 Report

The authors have responded to all my comments and the revised manuscript may be published. 

Reviewer 3 Report

Thank you for your comments. The answers were correct. I have no further questions.